# Pathogenesis of Chronic Kidney Disease Is Closely Bound up with Alzheimer’s Disease, Especially via the Renin-Angiotensin System

**DOI:** 10.3390/jcm12041459

**Published:** 2023-02-12

**Authors:** Ke Ma, Zi-Run Zheng, Yu Meng

**Affiliations:** 1The First Affiliated Hospital of Jinan University, Guangzhou 510000, China; 2Central Laboratory, The Fifth Affiliated Hospital of Jinan University, Heyuan 517000, China; 3Institute of Nephrology, Jinan University, Guangzhou 510000, China

**Keywords:** renin-angiotensin system, chronic kidney disease, Alzheimer’s disease

## Abstract

Chronic kidney disease (CKD) is a clinical syndrome secondary to the definitive change in function and structure of the kidney, which is characterized by its irreversibility and slow and progressive evolution. Alzheimer’s disease (AD) is characterized by the extracellular accumulation of misfolded β-amyloid (Aβ) proteins into senile plaques and the formation of neurofibrillary tangles (NFTs) containing hyperphosphorylated tau. In the aging population, CKD and AD are growing problems. CKD patients are prone to cognitive decline and AD. However, the connection between CKD and AD is still unclear. In this review, we take the lead in showing that the development of the pathophysiology of CKD may also cause or exacerbate AD, especially the renin-angiotensin system (RAS). In vivo studies had already shown that the increased expression of angiotensin-converting enzyme (ACE) produces a positive effect in aggravating AD, but ACE inhibitors (ACEIs) have protective effects against AD. Among the possible association of risk factors in CKD and AD, we mainly discuss the RAS in the systemic circulation and the brain.

## 1. Introduction

Alzheimer’s disease (AD), a neurodegenerative disease that occurs in old age and pre-senile age, is characterized by progressive cognitive impairment and behavioral impairment [1,2]. Epidemiological survey has estimated that AD comprised 60–80% of ~50 million dementia individuals around the world in 2018, and the number is projected to triple by 2050 [3]. The common risk factors for the development of AD are increased age [4], family history [5], degeneration or vascular dysfunction [6], obesity [7], hypotension or hypertension [8], diabetes [9], hyperlipidemia [10], and the existence of epsilon 4 allele of the apolipoprotein E gene (ApoE4) [4]. Kidney disease is recently proposed as a modifiable risk factor for AD. Studies showed that kidney disease patients expressed a higher level of amyloid precursor protein (APP), a key protein for protein-bound receptor sorting (SorLA). It acts as a central regulator of APP trafficking and processing and is expressed concurrently in neurons (cerebellum, hippocampus, and cortex), renal cells, and gene polymorphism, which is associated with late-onset AD [11,12,13]. Due to the limited effective pharmacotherapies of AD [14,15], it is more and more important to focus on studies based on the prevention against major and modifiable risk factors.

Chronic kidney disease (CKD) is defined in adult patients as a glomerular filtration rate (GFR) of less than 60 mL/min/1.73 m^2^, or a GFR of greater than 60 mL/min/1.73 m^2^ but with evidence of equal damage to kidney structures, and an onset time lasting for three months or more [16]. CKD is becoming more and more prevalent (10–13% of the population), irreversible, progressive, and associated with higher cardiovascular risk, stimulating the development of cognitive decline and the progression of AD [17,18,19,20]. Zhang et al. [21] reported that patients with AD in 50,550 CKD individuals were up to 10.8% and the rate has shown an upward tendency year by year. To date, the pathophysiology of AD and the role of CKD in AD progression are not completely clear. It is known that RAS significantly matters in the pathology of CKD [22,23]. Additionally, recent studies have shown that the RAS in the brain potentially contributes to dementia, and inhibitors of this system have been shown to be important, which demonstrated that RAS plays an important role in AD [24]. However, only a few studies have demonstrated the association between CKD and AD from the perspective of the RAS. Therefore, we mainly described the pathophysiology of AD and the role of CKD in AD progression from the aspect of RAS.

In this review, we briefly summarized the progression of the pathophysiology of CKD and AD. Secondly, we summarized the function of the RAS in the systemic circulation and brain. Thirdly, we demonstrated that the progression of the pathophysiology of CKD may be involved in the occurrence and the development of AD, especially from the perspective of the RAS, uremic toxins, erythropoietin (EPO), and extracellular vehicles (EVs). Finally, we discussed whether the pathogenic mechanism of the RAS in AD is associated with the pathogenic mechanism of the RAS in CKD. Understanding the potential mechanism between them may provide the potential possibility for RAS-related drugs to be used to prevent or treat AD.

## 2. Pathophysiology of Chronic Kidney Disease

Since 1990, the global incidence and prevalence of CKD and the number of associated deaths have increased by 89%, 87%, and 98%, respectively [25]. CKD is caused by abnormal development or injury, inflammation/immune mechanisms, or a toxic insult. Several pathways can perpetuate kidney damage by inducing the hyperfiltration and hypertrophy of remaining nephrons; and by promoting oxidative stress, inflammation, the accumulation of uremic toxins and EPO, vascular calcification, and RAS activation. The perturbation of the balance between pro-and anti-oxidant mechanisms results in metabolic dysregulation and the overproduction of reactive oxygen species (ROS) and/or oxidative end products of lipids, DNA, and proteins, thus leading to oxidative damage in cells, tissues, and organs [16,17,26,27]. The irreversible loss of normal renal tissue function and interstitial fibrosis causes pericytes to transdifferentiate into myofibroblasts, followed by massive extracellular matrix deposition and fibrosis and decreased EPO gene transcription [28,29,30]. The decline in renal function in CKD blocks the excretion of organic compounds, leading to the accumulation of uremic toxins in the body followed by chronic inflammation, endothelial dysfunction, damage to mitochondria, and oxidative stress [30,31,32,33,34,35,36,37,38,39]. The increase in proinflammatory cytokine production, oxidative stress, acidosis, and infections caused by persistent low-grade inflammation results in a dysregulated microvascular response to intrarenal regulators, leading to tubular damage, nephron shedding, and CKD onset [40,41,42,43,44,45,46,47,48]. The imbalance in circulation enhancers and vascular calcification inhibitors in CKD induces vascular calcification and osteogenic differentiation in vascular smooth muscle cells, as well as activating mediators of vascular calcification [47,49,50,51,52]. Glomerulosclerosis decreases downstream peritubular capillary blood flow, causing glomeruli in these areas to secrete excess renin; this further increases circulating angiotensin II (Ang II) levels, which then increases systemic vascular resistance and blood pressure and promotes sodium reabsorption in the proximal tubule and (via aldosterone) the collecting duct [53,54,55,56,57,58]. EVs could play an important role in the development of CKD [36,59]; patients with advanced CKD have elevated levels of platelet-, neutrophil-, erythrocyte-, and endothelial cell-derived EVs [60,61,62]. Platelet-derived EVs are procoagulant and prothrombotic [60,62], and an increased number of endothelial cell-derived EVs was shown to be correlated with endothelial dysfunction, atherosclerosis, and arterial stiffness [63]. The abovementioned processes can lead to alterations in podocytes and changes in glomerular structure and architecture, which can result in the sclerosis of nephrons, further deterioration of renal function, and fibrosis [64].

## 3. Alzheimer’s Disease Pathogenesis

As life expectancy increases and the global population ages, the prevalence of AD is expected to continue increasing (especially in developing countries), resulting in high disease and economic burdens. The number of people with dementia is projected to reach 152 million by 2050, with the largest increases in low- and middle-income countries [65,66]. Community surveys in China over the past few decades have found a marked increase in AD prevalence [67,68].

The most common cause of AD is the extracellular accumulation of misfolded Aβ proteins into senile plaques and the formation of NFTs containing hyperphosphorylated tau proteins [66,69,70]. Aβ proteins can cause various synaptic defects that have synaptogenic effects and can lead to neuronal death. Meanwhile, Aβ ligands can cause the dysregulation of Ca^2+^ homeostasis, mitochondrial damage, oxidative stress, altered axonal transport, and glial activation, further leading to synaptic damage and neurotoxicity. Aβ accumulation, in which tau phosphorylation plays a critical role, leads to NFT formation, cell loss, vascular damage, and dementia [71,72,73,74]. Tau oligomers at synapses are toxic to neurons and can cause synaptic damage before NFT formation, triggering neurodegeneration. At the same time, tau hyperphosphorylation causes conformational changes that enable the protein to sequester normal microtubule-associated tau; the resulting aggregates of paired helical filaments can cause synaptic dysfunction [71,72,75]. Both mechanisms may be linked to symptoms of AD including progressive memory loss, cognitive decline, and learning difficulties [76,77]. EVs contribute to the pathology of AD by spreading Aβ and tau, thereby promoting AD progression. Additionally, EVs are potential carriers of pathogenic AD-related proteins that impair neuronal function [78,79,80,81]. RAS activation in the brain; alterations in neurons, microglia, and astrocytes; neuroinflammation; vascular changes, and aging are all risk factors for AD (Figure 1).

## 4. Renin-Angiotensin System

CKD is closely associated with increased RAS activation [54,56,57,58]. In glomerulosclerosis, there is a decrease in downstream peritubular capillary blood flow [48], which causes glomeruli in these areas to secrete excess renin. This further increases the level of circulating Ang II, which not only has a direct vasoconstrictive effect of increasing systemic vascular resistance and blood pressure but also promotes sodium reabsorption in the proximal tubule and (through aldosterone) the collecting duct [23,53]. With fewer functional glomeruli in CKD patients, the GFR and perfusion pressure must be increased by higher systemic arterial pressure. The overactivation of the sympathetic nervous system in CKD stimulates renin production by renal juxtaglomerular cells, further aggravating kidney damage. In summary, the factors induced by CKD cause excessive activation of the RAS, leading to the upregulation of Ang II receptor type 1 (AT1R) [55]. The specific mechanisms involving the RAS are discussed in detail below.

### 4.1. RAS in the Systemic Circulation

The RAS is an important humoral regulatory system in the human body. The RAS functions in the circulation as well as in the blood vessel wall, heart, kidney, adrenal gland, and other tissues and it plays an important role in the normal development of the cardiovascular system, cardiovascular homeostasis, the maintenance of electrolyte and fluid balance, and the regulation of blood pressure [82,83].

Angiotensinogen (AGT) and renin are the core elements of the classical RAS pathway. AGT is synthesized in the liver, and renin is an aspartyl protease released by the juxtaglomerular cells of the kidney that cleaves AGT at the amino terminal. This generates the inactive decapeptide angiotensin I (Ang I), which is activated by ACE and hydrolyzed at the C-terminal His–Leu to yield the active octapeptide Ang II, the main effector of the RAS pathway. Ang II is cleaved at the N-terminal Asp residue into the heptapeptide angiotensin III (Ang III) by glutamyl aminopeptidase A(AP-A) but it also converted into Ang (1–7) by carboxypeptidase P-mediated cleavage at the Phe residue, by the monopeptidase ACE2, or the ACE cleavage of Phe–His from Ang (1–9). Ang III can be converted to the hexapeptide Ang IV by membrane alanyl aminopeptidase N, which cleaves the N-terminal Arg of Ang III. Ang IV can be further converted to Ang (3–7) by carboxypeptidase P and prolyl oligopeptides that cleave the Pro–Phe bond. Ang I is biologically inactive. Ang II and Ang III are full agonists at AT1R and AT2R and exert opposite effects. In contrast, Ang IV binds with low affinity to AT1R and AT2R but binds with high affinity and selectivity to AT4R [67,84,85,86,87,88,89,90,91,92].

Ang II has a variety of effects after binding to different Ang II receptors. The binding of Ang II and AT1R can induce a range of physiologic and pathologic effects such as inducing central sympathetic outflow, vasoconstriction, renal sodium reabsorption, and release of aldosterone and arginine vasopressin (AVP) from the adrenal and pituitary glands. Conversely, the binding of Ang II and AT2R can result in vasodilation, apoptosis, cellular proliferation, sympathetic inhibition, decreased sodium reabsorption, and the inhibition of AVP release [84,93,94]. AT1R has two subtypes but their distinct physiologic actions are not fully known. Ang III was shown to decrease sodium reabsorption and exert a cardioprotective effect after binding to AT2R [95]; and Ang IV can affect cognitive function, reduce neuronal apoptosis, and promote inflammation [96,97,98]. Ang 1–7 binds primarily to Mas receptors, counteracting many of the deleterious effects of Ang II [99].

### 4.2. RAS in the Brain

Unlike the systemic RAS, a local RAS is present in the heart, kidney, lung, liver, and retina [100,101]. However, there is little information available on the expression and regulation of the RAS in the brain [102]. The effects of the RASs in the central nervous system (CNS) were originally considered to be due to the activity of the circulating RAS components acting through the circumventricular organs on neurons to regulate blood pressure and sodium and water homeostasis in the brain [103]. However, using a variety of methodologic approaches including confocal laser microscopy, in situ hybridization, laser microdissection, and PCR, it was demonstrated that the brain has a local RAS independent of the peripheral RAS.

RAS components such as renin, angiotensinogen, ACE, Ang I, Ang II, and RAS-specific second messengers are expressed in the brain [68,104,105,106]. The most highly expressed component in neurons is the renin receptor ATPase H+ transporting accessory protein 2 (Atp6ap2). AGT is expressed in astrocytes, and AGT receptor (AGTR) expression overlaps with that of the Mas receptor (MasR) and Atp6ap2. The main AGTR subtype is expressed in microglia, which also express ACE2 and MasR. AGT, ACE, AGTR, Atp6ap2, and MasR are expressed in oligodendrocytes. There is increasing evidence that AT1R and AT2R are widely distributed in the CNS. Additionally, AGT constitutively produces many neuroactive peptides in astrocytes.

Renin cleaves AGT into Ang I in neurons and astrocytes. ACE converts Ang I into Ang II, which binds to AT1R and AT2R expressed by neurons, astrocytes, oligodendrocytes, and microglia in various brain regions [107]. ACE2 converts Ang II to Ang (1–7), which then binds to MasR. The activation of the Ang II/AT1R axis initiates a cascade of events that promotes oxidative stress, apoptosis, and neuroinflammation in several brain disorders. Angiotensin-II receptor blockers (ARBs), AT2R, and the activation of the ACE2/Ang (1–7)/MasR axis provide strong neuroprotection [108,109]. Ang II is converted to Ang III by aminopeptidase A/B, and then to Ang IV by aminopeptidase B, which can act on AT4R [110,111].

## 5. Association between CKD and AD

### 5.1. RAS Linking CKD and AD

Many components of the RAS system such as Ang II and Ang III are also expressed in the brain [112,113] and act via two mechanisms. The activation of the Ang II/AT1R axis initiates a series of events that promote oxidative stress, apoptosis, and neuroinflammation in several brain disorders. Moreover, ARBs, AT2R, and the activation of the ACE2/Ang (1–7)/MasR axis have neuroprotective effects (Figure 2) [111,113]. Glomerulosclerosis and the decrease in effective (perceived) blood flow in CKD patients result in excess renin release, an increase in circulating Ang II levels, and AT1R upregulation [6]. Ang II increases systemic vascular resistance and blood pressure and promotes sodium reabsorption in the proximal tubule and (via aldosterone) the collecting duct [114,115]. In a retrospective clinical cohort study, Douglas Barthold et al. [116] showed that in combination with statin and RAS-acting antihypertensives (AHTs), particularly ARB therapy, they may be more effective at reducing the risk of AD and related dementias. Michael Ouk et al. [117] reported that among ApoE4 non-carriers with AD, ARB use was related to the greater preservation of memory and attention/psychomotor processing speed, particularly compared with ACEI that do not cross the blood-brain-barrier (RR = 1.200, *p* = 0.003). More and more studies have suggested that RAS activation may be also involved in the development of AD, which is detailed with specific pathogenesis in the following sections (Table 1).

### 5.2. RAS and Aβ and Tau in AD

The overactivation of the RAS in the brain—especially Ang II-mediated signaling via AT1R—is associated with AD [135]. In one study, the cerebroventricular infusion of Ang II into aged normal rats increased both tau pathology and APP, leading to a rise in Aβ levels [136]. The link between ACE and Aβ and tau has been confirmed by evaluating the expression and distribution of RAS components in human postmortem brain tissue. ACE2 prevents pathogenic Aβ plaque formation. ACE2 enzyme activity was shown to be reduced by ~50% in the mid-frontal cortex of patients with AD compared with age-matched controls. Additionally, the ratio of ACE2 to ACE1 is decreased in AD patients [119]. In an animal model of sporadic AD, Ang (1–7) expression in the brain increased significantly with disease progression, and an inverse correlation was observed between the Ang (1–7) level and tau hyperphosphorylation. It is thought that components of the RAS aggravate AD by promoting extracellular Aβ deposition and enhancing the intracellular accumulation of pathologic tau protein [6,137]. However, the specific mechanism by which RAS activation links CKD and AD remains unknown.

### 5.3. RAS and Microglia

Inflammation, which encompasses a variety of immune reactions, is a prominent feature of AD. Microglia are the immune cells that mediate neuroinflammation [18], acting as brain-resident macrophages [138,139]. Microglia have two distinct phenotypes—namely, proinflammatory/classically activated (M1) and anti-inflammatory/alternatively activated (M2) phenotypes according to physiologic or pathologic conditions. The brain RAS plays an important role in microglia polarization. By binding to Ang II, AT1 stimulation triggers NADPH oxidase activation, leading to ROS generation and inducing a shift in microglia phenotype from M2 to M1 [140]. M1 microglia secrete proinflammatory cytokines that exacerbate cognitive dysfunction in the cortex, hippocampus, and basal ganglia by promoting ROS production and neuronal death [141]. The upregulation of AT1R in M1 microglia is associated with inflammation and cognitive impairment through a Toll-like receptor 4 (TLR4)-dependent mechanism [142]; thus, the AT1R-mediated activation of proinflammatory M1 microglia exacerbates inflammation, eventually causing AD [143]. Moreover, the activation of AT2R shifts microglia toward an M2 phenotype. AT2R and MasR activation decrease inducible nitric oxide synthase (iNOS) and the levels of inflammatory cytokines such as C-X-C motif chemokine ligand 12 (CXCL12), interleukin 1β (IL-1β), and IL-6 produced by M1 microglia while increasing anti-inflammatory markers such as IL-10 and IL-4 [143,144,145,146,147]. Thus, AT2R and MasR promote the activation of an M2 anti-inflammatory phenotype, which is a potential mechanism by which neuronal dysfunction and inflammation can be alleviated and cognition impairment can be reversed [143].

### 5.4. RAS and the Blood–Brain Barrier (BBB)

The BBB is a highly specialized endothelial cell membrane that regulates the entry of plasma-derived components, red blood cells, and leukocytes into the CNS and prevents the entry of potentially neurotoxic molecules [148,149,150]. In a clinical study, Michael Ouk et al. [117] reported that among APOE ε4 non-carriers with AD, ARB use was related to the greater preservation of memory and attention/psychomotor processing speed, particularly compared with ACEIs that do not cross the blood-brain-barrier (RR = 1.200, *p* = 0.003). Experiments in 5×FAD mice, which recapitulate the main features, and are a widely used model, of AD, showed that the BBB was disrupted in CKD by the action of Ang II [20,151,152], which also induced inflammatory and thrombotic phenotypes in the cerebral microcirculation. The binding of Ang II with AT1R results in leakage of the BBB and the entry of circulating toxins into the brain. Microglia are essential for CNS homeostasis and their overactivation leads to increases in NOS levels and the production of ROS and proinflammatory cytokines including tumor necrosis factor α (TNF-α), IL-1β, and IL-6 [153]. This causes neuronal injury and enhances BBB permeability via the activation of AT1R expressed by brain endothelial cells, resulting in a positive feedback effect.

In conclusion, the occurrence and development of AD are closely related to RAS activation, but it is unknown whether RAS activation in AD and CKD are related, and by which mechanism if so.

## 6. Uremic Toxins

Uremic toxins accumulate in the body in CKD, leading to neurotoxicity, BBB damage, ischemia/microvascular changes, neuroinflammation, and oxidative stress [154,155]. Patients with CKD have a similar risk of developing AD to the general population [2,156]. An accumulation of uremic toxins in the brain has been reported in patients with uremic syndrome: 5- to 20-fold (or more) increases in the levels of guanidine compounds have been detected in different brain areas [68,157]. Guanidine-based compounds are the main cause of cognitive deficits [157,158]; following kidney transplantation, the levels of these substances were found to decrease with a concomitant alleviation of cognitive deficits. The accumulation of guanidine compounds in the brain causes direct damage to neurons and astrocytes triggered by elevation in the levels of cytokines and interleukins, neuroinflammation, oxidative stress, and increased neuronal apoptosis [159,160,161]. Uremic toxins cause BBB damage [162,163], leading to vascular damage, the influx of endogenous and exogenous toxic chemicals and inflammatory substances into the brain, oxidative stress, neurotoxicity, and cognitive impairment [156].

## 7. Erythropoietin

Tubulointerstitial fibrosis in CKD patients leads to the loss of EPO secretion [6,18]. Both EPO and its receptor (EPOR) have been detected in the brain [164]. EPO/EPOR signaling is required for regular brain development and is essential for preventing neuron apoptosis, oxidative stress, and inflammation. The protective effects of EPO were first studied at the cellular level using hippocampal neurons and PC12 cells [130,165] and they involve the suppression of oxidative stress, the blockade of apoptosis, and tau phosphorylation induced by Aβ toxicity [131,132]. EPO was shown to alleviate Aβ-induced memory impairment and cognitive deficits by restoring vesicle release probability in Sprague–Dawley rats [132]. The mechanisms by which EPO protects against inflammation involve microglia activation, phosphatidylserine exposure, and protein kinase B (PKB) activity or the prevention of the release of proinflammatory cytokines including IL-6, TNF, and monocyte chemoattractant protein-1 (MCP-1). Thus, EPO is neuroprotective and may prevent or slow the progression of AD. In a large-scale clinical trial, EPO improved cognitive function and slowed progressive neuron atrophy in the brain [166,167,168]. These results suggest that the reduced secretion of EPO promotes cognitive impairment in patients with CKD.

## 8. Extracellular Vehicles

EVs have important physiologic functions and may contribute to the development and progression of inflammatory, vascular, malignant, infectious, and neurodegenerative diseases. In end-stage renal disease, circulating EVs were found to impair endothelial-dependent vasorelaxation, which was associated with a decrease in endothelial nitric oxide release and endothelial function. In a mouse model of indoxyl sulfate-induced CKD, intravenous administration of EVs from indoxyl sulfate-treated endothelial cells significantly reduced endothelial regeneration [80]. Additionally, animal and cell-based studies have shown that indoxyl sulfate homocysteine enhanced EV release in vitro and in vivo, leading to inflammation, apoptosis, cellular senescence, proliferation, calcification, and neointimal hyperplasia [80,169,170,171]. EVs from the brain of patients with AD contain elevated levels of Aβ oligomers and act as vehicles for the neuron-to-neuron transfer of these toxic species. Blocking the formation, secretion, or uptake of EVs was found to reduce both the spread of oligomers and associated toxicity [78,172]. Additionally, the overexpression of the AD-associated gene bridging integrator 1 (BIN1) stimulated the release of tau via EVs in vitro and aggravated tau pathology in PS19 mice [173]. Furthermore, the injection of physiologic levels of free-form tau into mice greatly diminished the propagation of tau protein compared with the injection of physiologic levels of EV-associated tau, suggesting that EVs are vehicles for the transfer of pathologic tau [174]. As mentioned above, EVs play a role in the pathogenesis of both CKD and AD; however, it is unknown whether CKD affects the pathogenesis of AD through cellular crosstalk mediated by EVs. This can be evaluated in future studies by injecting exosomes extracted from a CKD mouse model into the brain of normal or AD mice.

## 9. Endothelin

Endothelin (ET) plays a major role in the development of proteinuria, fibrosis, and CKD progression. ET-1 is involved in cell proliferation, hypertrophy, inflammation, and extracellular matrix accumulation, which attributes to the progression of CKD. With CKD progressions such as insulin resistance, dyslipidemia, ROS formation, and nitric oxide deficiency, the production of ET-1 will increase [175,176]. Hiddo J. L. Heerspink et al. [177] demonstrated that the selective ET-A receptor antagonist atrasentan could reduce albuminuria and reduce the risk of kidney failure in CKD patients [177]. Endothelin-converting enzyme inhibitors (ECEIs) have been reported to delay CKD progression by regulating autophagy, the NLRP3 inflammasome, and endoplasmic reticulum stress [178]. In addition, as a potent vasoconstrictor, ET-1 contributes to cerebrovascular dysfunction and neuroinflammation, which is associated with the progression of AD and related dementias [179,180]. Gulati et al. [181] showed that ET-B receptors agonist IRL-1620 reduced oxidative stress and improved learning and memory in an aged APP/PS1 transgenic mouse model of AD. Impaired clearance of Aβ has been considered one of the pathophysiologies of AD, while endothelin-converting enzyme (ECE)-1 and ECE-2 are known enzymes that could degrade Aβ in vivo. Studies have shown that alterations in ECE activity may be deemed as a cause for increased intraneuronal Aβ in AD [182,183]. To sum up, ECEI could delay the progression of CKD, while ECE could promote Aβ to improve AD. Whether we can explore the relationship between CKD and AD from the balance between ECE and ECEI, maybe a future research goal.

## 10. Conclusions and Future Perspectives

The available evidence suggests that CKD and AD are pathologically related through the RAS, uremic toxins, and EPO, which contribute to the occurrence and development of CKD and may aggravate the development of AD. In CKD, excess renin is released and increases circulating Ang II levels, resulting in AT1R upregulation and enhancing systemic vascular resistance, increasing blood pressure, and promoting sodium reabsorption in the proximal tubule and (through aldosterone) the collecting duct. In AD animal models, the cerebroventricular infusion of Ang II into aged normal rats increased both tau pathology and APP levels, leading to an increase in Aβ accumulation. It was also shown that Ang (1–7) expression in the brain increased with disease progression and that there was an inverse correlation between Ang (1–7) level and tau hyperphosphorylation. In AD model mice, Ang II not only impaired BBB function in the cerebral microcirculation but also induced inflammatory and thrombotic phenotypes. The binding of Ang II with AT1R damaged the BBB, leading to its leakage and the entry of circulating toxins into the brain. Additionally, AT2R and MasR promoted an M2 anti-inflammatory phenotype in microglia, which is a potential mechanism for alleviating neuronal dysfunction and inflammation and ultimately, for reversing cognition impairment. Based on the current evidence, we propose that the combination of Ang II and AT1R causes BBB leakage and activates microglia to secrete inflammatory factors that lead to apoptosis, neuronal injury, and neurodegeneration, resulting in the aggravation of AD; the activation of the AT2R/MasR axis produces the opposite physiological effect.

It remains unclear whether RAS imbalance in CKD is a cause of AD and vice versa. The following open questions warrant investigation in future studies: (1) Do CKD patients with AD have more severe imbalances in the RAS than those without AD? (2) What are the most significantly altered components of the RAS in CKD patients with AD, and are these components mainly proinflammatory (ACE/AT1R) or anti-inflammatory (ACE2/AT2R/MasR)? (3) Can the use of ACEI/ARB drugs prevent or delay the occurrence of AD? Answering these questions may provide insights that can guide the development of novel treatments for both diseases.

## Figures and Tables

**Figure 1 jcm-12-01459-f001:**
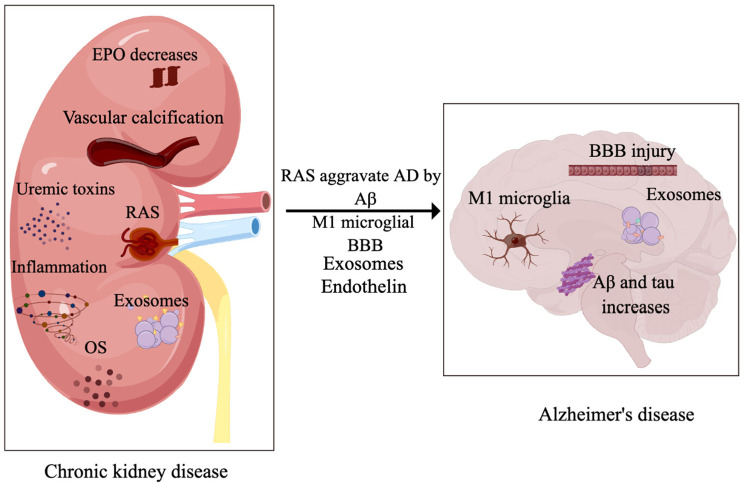
CKD and AD pathogenesis are closely related through the RAS.

**Figure 2 jcm-12-01459-f002:**
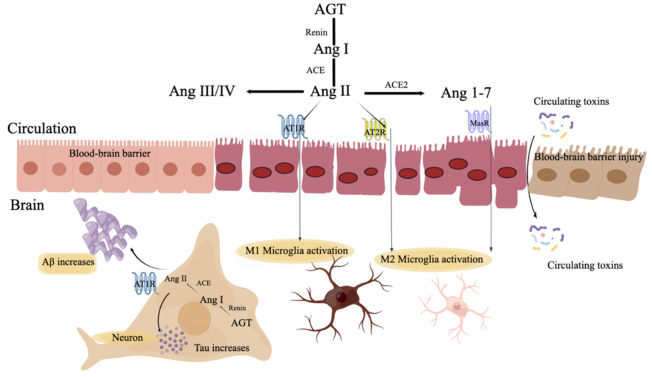
RAS involvement in AD. Components of the RAS bind with AT1R to cause injury to the BBB, resulting in increased cytokine release, the numbers of M1 microglia, and Aβ accumulation.

**Table 1 jcm-12-01459-t001:** The clinical studies that assess the association between CKD and AD.

	Factor	Attributes	Reference
Renin-Angiotensin System	ACE1	ACE1 accumulates in severe AD patients.	Miners et al. [118]
	ACE2	ACE2 activated, causes lower hippocampal Aβ and restored cognition	Kehoe et al. [119]
	ACEI	ACEI was higher in the temporal cortex of Alzheimer’s patients	Barnes et al. [120]
	ACEI	ACEI causes a slower rate of cognitive decline	Soto et al. [121]
	ARBs	ARBs significantly reduce the incidence and progression of Alzheimer’s disease and dementia	Li et al. [122]
	ARBs	ARBs rescue cerebrovascular and cognitive function in adults.	Ongali et al. [123]
	ARBs	ARBs could improve cognitive function, in particular immediate and delayed memory.	Fogari et al. [124]
	ACEI and ARBs	ARBs may have greater cognition protective effects than ACEI	Fournier et al. [125]
Uremic toxins	Uric Acid	Systemic hyperuricemia induces cognitive dysfunction	Lemma et al. [126]
	Indoxyl Sulfate	Chronic exposure to Indoxyl Sulfate leads to reduced locomotor activity and spatial memory, as well as increased stress sensitivity, and apathetic behavior	Karbowska et al. [127]
	Parathyroid hormone	Parathyroid hormone decline brain impairment by vitamin D	Larsson et al. [128]
Erythropoietin	EPO	Epo-EpoR significant cytoprotection by antioxidant, antiapoptotic, anti-inflammatory, neurotrophic, angiogenic, and synaptogenic activities.	Assaraf et al. [129] Viviani et al. [130] Ma et al. [131] Esmaeili Tazangi et al. [132]
Extracellular Vehicles	EVs	EVs contribute to peptide clearance from the extracellular space and reducing Aβ pathology	Yuyama et al. [133,134] Soares Martins et al. [80]

## Data Availability

Not applicable.

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
