# Peer review of "Pathogenesis of Chronic Kidney Disease Is Closely Bound up with Alzheimer’s Disease, Especially via the Renin-Angiotensin System"

_jcm, 2023, doi:10.3390/jcm12041459_

Round 1

Reviewer 1 Report

This is nice review manuscript. Manuscript is clearly written. I have few comments to be considered

1.       Suggest to avoid using abbreviation (CKD, AD) in the manuscript title

2.       Suggest to add table to summarize clinical studies that assess the association between CKD and AD

3.       In abstract, it was mentioned that ACE inhibitors have protective effect against AD. However, I did not see the discussion of clinical studies that investigate the effect of ACEI/ARB on AD or other impaired cognitive conditions.

Author Response

1.“Suggest to avoid using abbreviation (CKD, AD) in the manuscript title.”

Reply: Thanks for the suggestion. We have modified the manuscript title to “Pathogenesis of chronic kidney disease is closely bound up with Alzheimer´s disease, especially with the renin-angiotensin system”.

2.“Suggest to add table to summarize clinical studies that assess the association between CKD and AD”

Reply: Thanks for your comments. We already add Table 1 in the manuscript in lines 469 to sum up the clinical studies that appraise the associations between CKD and AD, including renin-angiotensin system, uremic toxins, erythropoietin and extracellular vehicles.

3.“In abstract, it was mentioned that ACE inhibitors have protective effect against AD. However, I did not see the discussion of clinical studies that investigate the effect of ACEI/ARB on AD or other impaired cognitive conditions.”

Reply: Thanks for your comments. We are sincerely sorry for the carelessness that ignored summarizing clinical studies about ACEI/ARB drugs. ACEI/ARB drugs that restore cognition in AD patients were the main reason we concentrate on the role of the renin-angiotensin system between CKD and AD. We add discussion in lines 277-285, lines 331-334, and Table 1 to list the clinical studies that confirm ACEI/ARB drugs can decrease cognitive decline and restore AD.

Looking forward to your reply.

Thanks and best regards,

Yu Meng

Reviewer 2 Report

Major comments:

1.    The text would greatly benefit from the reading by the native English speaker. Several sentences are difficult to be understood, especially in the abstract and Introduction. Moreover, there are many typo errors at the first page of the text.

2.    The text is rather superficial, without a detailed knowledge of the topic. Some introductory parts are very descriptive, covering more the pharmacology than the real information on the CKD and AD. Many references are referred to review articles instead of using primary papers. Several important systems were omitted at all, eg. endothelin system. The number of citation for a review article is quite low.

3.    The statement “…we elucidated that the progression” (line 38) is inappropriate and should be tempered.

Minor comments:

1.    The title needs rewording “Pathogenesis of chronic kidney disease is closely bound up with Alzheimer´s disease, especially with the renin-angiotensin system”.

2.    Titles of paragraphs should be self-explanatory, without abbreviations (eg. EPO, EVs). Abbreviations must be explained at the first place of their usage. A table with abbreviations would be helpful.

3.    Please use Alzheimer´s disease not Alzheimer´s Disease throughout the text.

4.    Latin words should be in italics (eg. in vivo, in vitro).

5.    Quadrat meter should be m2 not m2

Author Response

1.“The text would greatly benefit from the reading by the native English speaker. Several sentences are difficult to be understood, especially in the abstract and Introduction. Moreover, there are many typo errors at the first page of the text.”

Reply: We were really sorry for our careless mistakes. We have checked entire text carefully to make sure correctness. In addition, we not only rechecked the words and grammar, but also carried out  text editing. We are sincerely sorry for our negligence.

2.“The text is rather superficial, without a detailed knowledge of the topic. Some introductory parts are very descriptive, covering more the pharmacology than the real information on the CKD and AD. Many references are referred to review articles instead of using primary papers. Several important systems were omitted at all, eg. endothelin system. The number of citation for a review article is quite low.”

Reply: Thank you for your suggestion. According to your comments, we have made corresponding supplements. Based on our ability and the current academic frontier research, we have made a relatively fundamental summary. In terms of other mechanisms (such as endothelin system, etc.), our understanding is relatively limited, so we have not added much supplement. We will further discuss it in the next research. It is undeniable that the level of our article is limited. However, we are carrying out some valuable experiments for the relevant content. This review is only the main part of the relevant information we can obtain.

3.“The statement “…we elucidated that the progression” (line 38) is inappropriate and should be tempered.”

Reply: Thanks again for your valuable suggestion. We have carefully considered the content of the article and the previous statement is inaccurate. Therefore, the adjustment has been made, and the adjustment content is clearly visible in the article (line 59).

4.“The title needs rewording “Pathogenesis of chronic kidney disease is closely bound up with Alzheimer´s disease, especially with the renin-angiotensin system”.”

Reply: Thanks for your suggestion. We have adjusted the title to “Pathogenesis of chronic kidney disease is closely bound up with Alzheimer´s disease, especially with the renin-angiotensin system”

5.“Titles of paragraphs should be self-explanatory, without abbreviations (eg. EPO, EVs). Abbreviations must be explained at the first place of their usage. A table with abbreviations would be helpful.”

Reply: Thanks for your careful checks. We are sorry for our carelessness. According to your suggestion, we have corrected them in the manuscript. We modify the titles of paragraphs “EPO” to “Erythropoietin”, and “EVs” to “Extracellular Vehicles”. And we added abbreviations of the words in the manuscript in lines 502-591.

6.“Please use Alzheimer´s disease not Alzheimer´s Disease throughout the text.”

Reply: Thanks for your careful checks. We are sorry for our carelessness. According to your suggestion, we have corrected them in the manuscript.

7.“Latin words should be in italics (eg. in vivo, in vitro).”

Reply: Thanks for your careful checks. We are sorry for our carelessness. According to your suggestion, we have corrected them in the manuscript.

8.“Quadrat meter should be m2 not m2”

Reply: Thanks for your careful checks. We are sorry for our carelessness. According to your suggestion, we have corrected them in the manuscript.

Looking forward to your reply.

Thanks and best regards,

Yu Meng

Reviewer 3 Report

Ma et al, reviewed relationship between CKD and AD with recent published papers. Because patients of CKD have had cognitive decline. Thus, Authors focus on these two topics.

Each topic is clearly understandable in this manuscript. However, I am not clear that why do authors focus on only AD? Because there are many disorders of dementia in the aged people. Tauopathy, one of the dementias, occurs without Abeta deposition in the brain. Therefore, authors should write the reason of focusing only AD.

Please say about this reason.

Author Response

1.“Ma et al, reviewed relationship between CKD and AD with recent published papers. Because patients of CKD have had cognitive decline. Thus, Authors focus on these two topics. Each topic is clearly understandable in this manuscript. However, I am not clear that why do authors focus on only AD? Because there are many disorders of dementia in the aged people. Tauopathy, one of the dementias, occurs without Abeta deposition in the brain. Therefore, authors should write the reason of focusing only AD. Please say about this reason.”

Reply: Thank you for your suggestion. The corresponding supplements have be added in the manuscript (lines 28-40). The incidence rate of Alzheimer's disease is increasing and closely related to a variety of common risk factors. The raise of typical phenotypic protein of Alzheimer's disease can be seen in patients with kidney disease, so we consider that chronic kidney disease is closely related to Alzheimer's disease.

Looking forward to your reply.

Thanks and best regards,

Yu Meng

Round 2

Reviewer 1 Report

all of my comments were addressed properly.

Author Response

Many thanks for the information concerning our manuscript submitted to Journal of Clinical Medicine

Reviewer 2 Report

The authors substantially improved the quality of the paper, so it could be published in the present form.

Author Response

(The authors gave the same response as above.)
